# Assessing Indoor Climate Control in a Water-Pad System for Small-Scale Agriculture in Taiwan: A CFD Study on Fan Modes

**DOI:** 10.3390/bioengineering10040452

**Published:** 2023-04-07

**Authors:** Jia-Kun Chen, Yung-Ling Sun, Chia-Chi Hsu, Tzu-I Tseng, Yu-Chuan Liang

**Affiliations:** 1Institute of Environmental and Occupational Health Sciences, College of Public Health, National Taiwan University, No. 17, Xu-Zhou Road, Taipei 100, Taiwan; kandayu110618@gmail.com (Y.-L.S.); asd28474840@gmail.com (C.-C.H.); 2National Center for High-Performance Computing, National Applied Research Laboratories, No. 7, R&D 6th Rd., Hsinchu Science Park, Hsinchu 300, Taiwan; titseng@narlabs.org.tw; 3Agricultural Biotechnology Research Center, Academia Sinica, No. 128, Sec. 2, Academia Road, Nankang, Taipei 115, Taiwan; ycliang@sinica.edu.tw

**Keywords:** layer house, fan arrangement, family farming, temperature–humidity index, heat stress

## Abstract

Heat stress poses a significant challenge to egg production in layer hens. High temperatures can disrupt the physiological functions of these birds, leading to reduced egg production and lower egg quality. This study evaluated the microclimate of laying hen houses using different management systems to determine the impact of heat stress on productivity and hen health. The results showed that the ALPS system, which manages the hen feeding environment, effectively improved productivity and decreased the daily death rate. In the traditional layer house, the daily death rate decreased by 0.045%, ranging from 0.086% to 0.041%, while the daily production rate increased by 3.51%, ranging from 69.73% to 73.24%. On the other hand, in a water-pad layer house, the daily death rate decreased by 0.033%, ranging from 0.082% to 0.049%, while the daily production rate increased by 21.3%, ranging from 70.8% to 92.1%. The simplified hen model helped design the indoor microclimate of commercial layer houses. The average difference in the model was about 4.4%. The study also demonstrated that fan models lowered the house’s average temperature and reduced the impact of heat stress on hen health and egg production. Findings indicate the need to control the humidity of inlet air to regulate temperature and humidity, and suggest that Model 3 is an energy-saving and intelligent solution for small-scale agriculture. The humidity of the inlet air affects the temperature experienced by the hens. The THI drops to the alert zone (70–75) when humidity is below 70%. In subtropical regions, we consider it necessary to control the humidity of the inlet air.

## 1. Introduction

Heat stress is a significant issue that affects egg production in layer hens. When layer hens are exposed to high temperatures, their physiological functions are disrupted, which can decrease egg production and quality. This, in turn, results in significant economic losses to poultry farmers. Several factors can affect the quality of eggs, and heat stress is just one of them. Previous studies have shown that the health conditions of layer hens, the thermal environment they are in, the microbial biomass of the breeding environment, and the type of cages they are housed in can all impact egg quality. The thermal environment is a crucial factor that affects the overall health and productivity of poultry. Poultry animals are highly sensitive to changes in temperature and humidity levels, and the thermal environment can impact various aspects of their physiology and behavior. Previous studies have shown that the quality of eggs is affected by multiple factors, such as the health conditions of hens, the thermal environment, the microbial biomass of the breeding environment, and the type of cage. Among them, the thermal environment is the key main factor [1,2,3,4]. It can impact the growth rate, appetite, feed utilization, egg production rate, and the quality of meat and eggs in poultry [5]. The existing literature has demonstrated that the heat resistance of chickens could be enhanced through early heat regulation or feed restriction, while the heat resistance of broilers can be improved in the short term [6]. The thermal environment conditions, including temperature and wind speed, are critical for animal production. These factors can impact animal welfare, the chicken’s thermoregulation system, and the production response [3]. However, different species have different ways of determining thermal comfort [7]. People used many methods to examine the environment’s temperature–humidity index of chickens [8]. Du et al. [9] used the sound of chickens to evaluate whether the hens were in a stressful situation or not. Perera et al. [10] used temperature and humidity to assess the growth performance of hens.

Due to economic considerations, the traditional battery cage is the primary type of cage used in Taiwan. However, the high temperature can cause heat stress, which can negatively affect the hens and result in poor egg production [11]. Animal welfare has become an increasingly important consideration in recent times. Furnished cages have been introduced for hens, and the tunnel-type layer house with a water-pad system has become the dominant approach. In the context of globalization and climate change, agriculture is facing a critical moment of transformation, which needs to be utilized to stabilize the economy and meet the livelihood needs of people. In Taiwan, with its unique geographical and cultural background, family farming is an integral part of the agricultural ecology. Taiwan’s agricultural economy is predominantly based on small-scale farming. The defining feature of this system is that many farming households own small plots of land and focus on cultivating crops such as fruits, vegetables, rice, and flowers. This model has many advantages, including the promotion of rural employment and a higher income for farmers. It also helps to maintain the efficiency of land use, improve product quality, protect the environment, and preserve rural culture and community cohesion. However, small-scale farming also faces many challenges, such as a lack of resources, insufficient technological advancements, and market instability. Therefore, Taiwan needs to implement appropriate policies to support small-scale farming, improve agricultural productivity and competitiveness, and promote sustainable agricultural development. The layer house with a water-pad system is an essential part of the transformation of family farming. Family farming in Taiwan is often characterized by small-scale agriculture, where the farm is typically passed down from generation to generation, and the family’s livelihood is closely tied to the farm’s success. As a result, the development of the layer house with a water-pad system is gaining traction in Taiwan’s small-scale agriculture. However, given Taiwan’s subtropical climate, the design of the layer house with a water-pad system needs to be adjusted to meet local climate conditions. Despite its many advantages, the design of the layer house with a water-pad system needs to be tailored to suit Taiwan’s subtropical climate. The weather in Taiwan is hot and humid for most of the year, which can make it challenging to maintain optimal temperature and humidity levels inside the layer house.

Previous studies have highlighted the importance of an appropriate air temperature, humidity, air velocity, and air quality for optimal hen performance. When ambient temperatures are too high or low, a mechanical ventilation system with cooling and heating capabilities is necessary. Moreover, the ventilation system should be adjusted according to the season. The water-pad ventilation system, also known as the tunnel ventilation system, utilizes mechanical ventilation to regulate the indoor temperature of the layer house in response to seasonal changes [12,13,14]. The water-pad system is an effective way to reduce the indoor ambient temperature of laying hens. However, the system generates water vapor during operation, which can increase the humidity of the environment. In Taiwan’s high-temperature and high-humidity environment, high humidity can exacerbate thermal stress and negatively affect the health of layer hens. For operators, managing the high-temperature and high-humidity environment has always been a significant challenge. Therefore, it is essential to carefully consider indoor climate control in water-pad layer houses.

To develop a water-pad system suitable for small-scale agriculture in Taiwan, this study utilized computational fluid dynamics (CFD) to investigate the impact of fan mode on indoor climate control in the water-pad system. A newly constructed layer house at a university experimental farm was chosen as the site for examining the influence of fan mode on indoor temperature distribution. Additionally, we selected two commercial laying hen houses and integrated our self-designed intelligent system into their management system to examine the effect of the intelligent system on the management mode of small-scale agriculture.

## 2. Materials and Methods

We developed an Artificial Intelligence Layer Production System (ALPS) to manage farms and installed it in commercial laying houses in Taiwan for intervention. ALPS helps us collect information on the indoor climate in layer houses to validate our numerical computations.

Our investigation focused on the effect of humidity and three different fan-operation modes on temperature distribution in a layer house at a university in Taichung, Taiwan. We conducted CFD validation, analyzed the flow of three fan arrangement models, and calculated the Temperature–Humidity Index (THI).

First, we built a CFD model based on information collected on-site. We then used the CFD model to calculate the results for the three fan modes. Finally, we compared the temperature distributions of the three models and calculated the THI, taking into account the humidity, to estimate the actual temperature experienced by the hens. 

### 2.1. Artificial Intelligence Layer Production System (ALPS)

In this study, we developed a preliminary version of an Artificial Intelligence Layer Production System (ALPS), as depicted in Figure 1a. ALPS utilizes IoT technology to regulate the indoor climate and environmental conditions of livestock houses. It enables control over various factors such as fan speed, temperature, humidity, ammonia and carbon dioxide concentration, the sprinkler system, intake door switch, and a small weather station. In the future, we plan to add biological data, such as animal body temperature, food intake, and the distribution of bacterial colonies in the pasture. Figure 1b illustrates the ALPS interface, which includes the indoor environment, layer information, water pump, curtain and light control, and fan control. The administrator can easily manage the indoor microclimate system through a mobile phone, tablet, or computer using the Internet.

### 2.2. On-Site Measurement at Layer House

The diagram in Figure 2 displays the location of measuring points within a 100 m long water-pad layer house in Chiayi. Our study focused on monitoring the temperature and humidity levels within the cages.

Figure 3 depicts a closed water-pad layer house in Taichung, measuring 24 m in length, 7.3 m in width, and 3.6 m in height. The layer house contained two rows of four-layer welfare chicken cages. At the front end of the layer house, there was a water-pad system cooling system with a width of 2 m on both sides and a height of 3 m. The rear of the layer house had six 36-inch fans installed. Figure 3 shows the dimension drawing of the layer house. 

Figure 4a shows the water-pad system’s dimension drawing, while Figure 4b shows the fan’s dimension drawing. Figure 4c indicates the measurement points SP1-SP5’s locations. In the layer house, SP1 and SP4 were installed 7.2 m from the water-pad system, SP3 was installed 12.3 m away, and SP2 and SP5 were installed 22.7 m away. The measuring points’ heights were 2 m and 2.7 m. An environmental monitoring system consisting of CO_2_, temperature, and humidity sensors (AVC-310, Aecl Group, New Taipei City, Taiwan) was installed in the house to monitor the environment.

Figure 5 shows the cage group classification and arrangement of hens in the layer house. We separated the area of the cage into seven groups. G1 to G7 were the number groups of the cages. Figure 5a shows that we divided cages into seven groups in the middle of the layer house space from the water-pad system to the fan. Figure 5b illustrates the distance between the hens in the cages. We made arrangements for the hens to be dispersed in a welfare cage, as shown in Figure 5b. Each section had four zones from up to down, where we kept the hens. There were 30 layers in each cage, with a total of 1680 layers. We simplified the hen geometry to be a rectangular block of 17.5 m × 11.5 m × 10.5 m. For the arrangement of 6 fans, we had three modes, as shown in Figure 6. As seen in Figure 6, gray fans were on, and white fans were off.

### 2.3. Body Surface Temperature Measurement of Hens

The microchip Bio-ThermTM (Destron Fearing^TM^, DFW Airport, TX, USA) was implanted into the subcutaneous tissue of the nape of 36 hens, and a scanner read the temperature measured by the chip. The temperature recording period was from July 2021 to December 2021. In total, 46 pieces of temperature data were obtained for each hen. The measurement results are shown in Figure 7. The lowest temperature was 38.1 °C, the highest temperature was 42.3 °C, and the average temperature of the 36 chickens was 40.7 °C.

### 2.4. Governing Equations

Computational fluid dynamics (CFD) was used to analyze the flow in space. Equation (1) is the equation of mass conservation.
(1)$$\nabla \cdot \left(\rho \overset{⃑}{V}\right)=0$$

Generally speaking, the airflow in the spaces is incompressible flow. Therefore, the density of the air in the spaces is constant. Therefore, we modify Equation (1) to (2).
(2)$$\nabla \cdot \overset{⃑}{V}=0$$

Equation (3) is the steady-state momentum equation called the Navier–Stokes equation.
(3)$$\rho \left(\overset{⃑}{V}\cdot \nabla \right)\overset{⃑}{V}=-\nabla p+\mu \nabla^{2}\overset{⃑}{V}$$

Equation (4) is the steady-state equation of energy conservation.
(4)$$\rho c_{p}\left(\overset{⃑}{V}\cdot \nabla T\right)=k\nabla^{2}T$$

### 2.5. Boundary Conditions

Boundary conditions were set to solve the Equations (2)–(4) in the study. The boundary conditions were set as follows. The water-pad system at the front of the layer house was set to *T*_wp_ = 23 °C. The temperature *T*_layer_ was set to 40 °C (the body-surface temperature around the layer) according to the measurement of 36 layers. The velocity of the fans, *v*, was set to 2, 4, and 6 m/s. Other boundaries, such as the walls, roofs, etc., in the house were set as wall boundaries, and the wall temperature *T*_wall_ = 25 °C.

Water pad inlet: Set the front-end water curtain of the hen house as a pressure outlet boundary condition during the simulation, and set the outlet temperature (*T*_wp_) to 23 °C.
(5)$$\mathrm{Water}\;\mathrm{pad}:\;\left\{\begin{array}{l} u=0\\ v=0\\ w=0\\ T_{wp}=23\;{}^{\circ}C \end{array}\right.$$

Layers: Set the hen flock as a heat source, so that the heat dissipation of the layers mainly comes from the surface of the layer and the heat generated by respiration. The specific heat dissipation value is affected by the environmental temperature and the physical condition of the layer. The temperature *T*_layer_ is set to 40 °C (layer surface temperature).
(6)$$\mathrm{Layer}:\;\left\{\begin{array}{l} u=0\\ v=0\\ w=0\\ T_{layer}=40\;{}^{\circ}C \end{array}\right.$$

Fan: The fan outlet is a velocity outlet with temperature *T*_f_ set to 30 °C.
(7)$$\mathrm{Fan}:\;\left\{\begin{array}{l} u=0\\ v=0\\ w=v\\ T_{f}=30\;{}^{\circ}C \end{array}\right.$$

Other boundaries: Set the walls and roof inside the house as wall boundaries, and ignore the evaporation of water vapor on the floor and water tank inside the house. Set the wall temperature *T*_wall_ to 25 °C.
(8)$$\mathrm{Wall}:\;\left\{\begin{array}{l} u=0\\ v=0\\ w=0\\ T_{wall}=25\;{}^{\circ}C \end{array}\right.$$

### 2.6. Numerical Simulation

This study used commercial software Ansys Fluent R19 (ANSYS, Inc., Pittsburgh, CA, USA) and SolidWorks Flow Simulation (Dassault Systèmes SolidWorks Corporation, Conker County, MA, USA) as the CFD tools. For this study, the flow model was set to a three-dimensional, fully turbulent, and incompressible flow model under the assumptions of RANS (Reynolds-averaged Navier–Stokes equations). The CFD tools use a semi-implicit method to handle the pressure-linked equations in the velocity-pressure coupling algorithm, and employ a second-order upwind discretization. In the turbulence model, the study used the standard k-ε turbulence model to model the turbulence characteristics of the fluid. The standard k-ε model is the most commonly used model for simulating turbulence in computational fluid dynamics [15], and is a two-equation turbulence model that provides a closure for the Reynolds-averaged Navier–Stokes (RANS) equations. It is based on the assumption that the turbulent kinetic energy (k) and its dissipation rate (ε) are the most important variables in the turbulent flow, and that they are transported separately from the mean flow variables. The k-ε model had been recognized in the field of indoor ventilation research [16,17,18]. 

The mesh of the model was shown in Figure 8. This study simulated a layer house with a capacity of 20,000 birds using seven different grid numbers: 1,140,650, 2,429,069, 3,364,980, 7,007,939, 13,835,801, 16,035,711, and 19,753,187. In each of the five rows of the layer house, three monitoring points were selected at the center of the aisle, with *x* = 5 m, 50 m, and 95 m, respectively. A total of 15 monitoring points were used to obtain wind speed data as independent references for each grid size. The grid independence results showed that for grid sizes above 13,835,801, the values of the 15 monitoring points reached a convergence criterion of 10^−3^. The grid independence was shown in Figure 9.

### 2.7. Temperature–Humidity Index (THI)

The temperature–humidity index (THI) was used to assess the impact of environmental conditions on the thermoregulatory mechanisms of animals [19,20,21].
(9)$$THI=0.6\times T_{db}+0.4\times T_{wb}$$

Zulovich and DeShazer [20] produced THI charts based on surveys of laying hens and physiological responses. They classified THI into four levels: comfort (THI < 70), alert (THI: 70–75), dangerous (THI: 76–81), and emergency (THI > 81) areas.

## 3. Results

### 3.1. ALPS System in Field

In Table 1 shows that the age of the hens ranged between 132 weeks and 136 weeks. Table 1 shows that the daily death rate ranged between 0.086% and 0.041%, decreasing by 0.045%. The daily production rate ranged between 69.73% and 73.24%, increasing by 3.51%. The operators said that they used the ALPS system to manage the hen feeding environment, and that it effectively improved the impact of heat stress caused by traditional feeding methods on laying hens and increased productivity and profits. This farm uses battery cages as the breeding space, which is a typical traditional breeding method in Taiwan. They usually put 3–4 hens in the same battery cage. However, this feeding method can have health impacts on the hens, and heat stress especially can cause heat-related illness in the hens, which can affect egg production and quality.

Table 2 shows that the age of hens ranged between 50 weeks and 54 weeks. Table 2 shows that the daily death rate ranged between 0.082% and 0.049%, decreasing by 0.033%. The daily production rate ranged between 70.8% and 92.1%, increasing by 21.3%. This farm uses a commercial water-pad system with a simple automated microclimate control system to ensure the quality of the environment inside the house. 

In Figure 10, we found that in the design of the water-pad system, the temperature increases toward the end of the room, especially after more than 30 m (the red dash-line labeled in Figure 10), and the temperature increases linearly. We deduced that the main reason for this is that the structure causes the temperature to accumulate at the end of the room. The water-pad structure is a tunnel structure, and the temperature of the hens at the front is transferred to the back, so the hens in the back have to bear the temperature from the front and the temperature of their environment.

### 3.2. Simplified Model Validation 

Table 3 shows the temperature measured in the layer house, the temperature calculated by CFD, and the difference between them. The difference between the measured temperature and the temperature of the CFD model ranged between approximately 0.5% and 10.3%. The average difference was about 4.4%. We found data with higher variance at SP2, SP3, and SP5. We inferred that the reason for these differences is the motors and lights on site. Around SP2, SP3, and SP5, the temperature produced by the motor and the lamp affected the measurement to produce these differences. Table 1 shows that the average error between the CFD and measurement was 4.4%, which is an acceptable range in the complex environmental conditions of laying hens. 

In this study, we established a simplified hen geometry that was dominated by rectangular block geometry. The simplified model helped to build many hen models for designing the indoor microclimate of commercial layer houses when we were in the pre-design evaluation stage. Most commercial layer houses house tens of thousands of hens. Thus, building the geometry according to the accurate scale of each hen would lead to excessive calculation and consume a massive amount of resources in terms of time and cost. This is true even in today’s era and the rapid development of various software and hardware. 

The problem for designers is establishing bird geometry when evaluating the internal microclimate of a water-pad system house. Indeed, they often encounter increased computational costs due to excessive hen geometry. Some researchers therefore use porous media to build models of the chickens and coops together [22,23]. When researchers use a porous model for calculations, there are several essential points to be aware of. 

Firstly, the pressure difference must be tested in all directions under different conditions, secondly, the porous model should use differential pressure as the driving source for fluid flow, and lastly, researchers need to set the model’s pressure difference in all directions. 

In addition, the media distribution of the porous model should be assumed. The porous media model assumes that the internal media is uniformly distributed; that is, the porosity is uniformly distributed.

However, in reality, hens are not evenly distributed in the cage, and the resulting pressure differential cannot be accurately assessed. The simplified hen model in this study was devised between complex geometry and the porous model, which helped reduce the duration of the layer house design process, making it thus more convenient for commercial applications.

### 3.3. Fan Models vs. Heat Accumulation in Laying House

Figure 11 shows the 2D temperature distribution in the cage area of the house in the three models. Table 4 lists the temperature distribution in the cage area of the house in the three models. All three models showed that the temperature at the tail end increases. When the fan increased the velocity, the average temperature in the house was lowered in all three models. The largest front-to-rear temperature difference occurred at different wind speeds in Model 2. The smallest temperature difference was of about 0.4 °C when we used Model 3 and the fan velocity was set to 6 m/s, as shown in Table 2. The airflow temperature received by the hens in G1 was the temperature after the water-pad system. From G2, the temperature in this zone was the ambient temperature of the hens in the previous zone. The temperature inside the cage built up to the back end. The temperature of the rear end rose because the fan had resistance, and the fluid had no way to go out the first time. The hens at the back end were in a higher ambient temperature than those at the front end. The heat stress of high temperatures affected hen health and egg production [24]. 

The velocity distribution in the house affected the temperature distribution. Figure 12 shows the air velocity contour inside the house. In Model 2, the four fans were located in the center aisle. The wind speed in the central aisle was more significant than in the aisles on both sides. Therefore, the temperature in the cage area of Model 2 was higher than that of the other two models. Figure 12c shows that the air inside Model 3 was forced out by the four fans below. The indoor temperature difference in Model 3 was similar to Model 1. However, Model 3 had only four fans, which helped the farm operators save energy and reduce operating costs.

### 3.4. Effect of Outdoor Air Humidity on Hen Body Temperature

Table 5 lists THI in different areas, showing that the humidity ranged between approximately 80 and 100%, as detected by 24 h monitoring on-site. The average THI was approximately in the range of 75.5–77.6 °C. All three models were in the dangerous zone, and the THI of Model 2 was the highest among the three models. 

In Table 6, we observe that a drop in humidity can bring the THI down to the alert zone. When the humidity is below 70%, THI drops to the alert zone. It requires pre-processing of the incoming air’s humidity, which contributes to improved hen welfare and egg quality. THI consists of temperature and humidity. Changes in humidity affect changes in the overall THI. The body temperature, a combination of humidity and temperature, is the body temperature felt by the hen if the humidity of the air entering the house is controlled below 70% to reduce the THI.

## 4. Discussion

This study mainly focuses on the indoor microclimate model of a water-pad layer house, which is suitable for small-scale agriculture. 

Taiwan’s climate is affected by climate change, and temperature changes are becoming more and more obvious. In this study, we found that the modern water-pad layer house also has the problem of a high-temperature environment, and that hens are therefore still affected by heat stress in the house. Heat stress refers to the pressure placed on living organisms in high-temperature and high-humidity environments. High temperature and humidity hurt laying hens’ productivity in the production of eggs. The perceived temperature generated by temperature and humidity is a component of heat stress assessment. As temperature and humidity increase, the perceived heat sensation of laying hens also increases, affecting their productivity. Therefore, to maintain the productivity and health status of laying hens, it is necessary to monitor and regulate the temperature and humidity in the environment, keeping them within a range that the hens can adapt to [25]. The traditional feeding method leads to an environment that is not conducive to the growth of the hen and the production of eggs; this is because the traditional feeding mode is to let the living environment of the hen mainly rely on natural ventilation. This method is not conducive to large-scale breeding.

Therefore, the modern feeding method uses a water-pad layer house, mainly using mechanical ventilation as the management mode of indoor microclimate. We have always thought that the water-pad feeding model has improved the environmental control problems in the traditional feeding model. However, the biggest problem with the structure of the water-pad model is that temperature accumulation occurs at the outlet of the layer house fan.

In this study, we found that the water-pad system does not entirely solve the problem of thermal stress. Heat accumulation is more apparent when the size of the water-pad layer house is more extensive [26,27]. In this study, we also found that heat accumulation occurs in the water-pad system, and the temperature rises when the structure length exceeds 30 m. The heat accumulation is mainly the result of the tunnel structure, which is the water-pad system.

The airflow flows from the front to the rear in the tunnel structure, which causes the temperature to pass from the front to the rear. Therefore, the temperature in the rear area is affected by the temperature in the front and then rises. At the same time, the fan at the end has an impedance. When a large amount of airflow passes through a fan with a fixed area, the airflow cannot flow out immediately due to the geometrical influence of the fixed area, which is also one of the reasons for temperature accumulation.

To further understand the cause of heat accumulation, we used a simplified computational fluid dynamics (CFD) model of the tunnel structure and conducted field measurements. We took this a step further in order to improve heat build-up and adjusted the fan modes. The results showed that using only the lower exhaust fan (Model 3) was an effective fan mode in this study. However, our findings indicated that water-pad systems might not effectively reduce the thermal stress on animals in areas with high humidity. When the outdoor humidity exceeds 70%, the indoor temperature–humidity index (THI) falls into a dangerous zone. The higher the humidity, the less heat that can be dissipated through evaporation, resulting in a higher perceived temperature for the animals. This is particularly relevant in Taiwan, a country that has a subtropical climate with high temperatures and high humidity; this significantly impacts the heat stress on animals subjected to indoor breeding methods using water-pad structures.

## 5. Conclusions

The novelty of this study is the use of computational fluid dynamics (CFD) to investigate the impact of the fan mode on indoor climate control in the water-pad system. This represents a new approach to studying the indoor climate of the water-pad system and provides insights into how to optimize the system for improved temperature distribution.

We came to the following conclusions for indoor microclimate control in layer houses:

The CFD of the simplified model can quickly obtain the assessment of indoor climate control. We verified the calculation results of the CFD model through on-site measurements, and the average error was controlled at 4.4%. We verified that the results show that the hen shape can be simplified to obtain results close to the actual measurement.

The temperature in the rear was higher than in the front, an inherent problem in designing the tunnel-type water-pad system layer house. Because the temperature of the hens in the front was transferred to the rear due to the airflow, the hens in the back were exposed to a higher temperature than those in the front. In addition, the air in the rear could not be discharged quickly for the first time due to the fan unit’s resistance, so the rear temperature was in a high-temperature state. According to the results of this study, the length of the layer house was controlled within 30 m, and the Model 3 design was used to control the temperature difference between the front and rear to 0.4 °C.

The apparent temperature of the hens was increased due to increased inlet humidity. The THI was located in a dangerous zone when the inlet air humidity was over 80%. When the humidity is below 70%, THI drops to the alert zone. The humidity of the inlet air affected the temperature distribution throughout the house. In subtropical regions, we considered it necessary to control the humidity of the inlet air. This way, the house’s temperature and humidity could be controlled entirely.

This study analyzed a layer house with a water-pad system with three fan modes. Based on the conditions of Taiwan’s average relative humidity, which is about 60–80%, and the needs of small-scale agriculture, the research results suggest that Model 3 should be used as the operation mode of layer houses with a water-pad system in small-scale agriculture. It provides small-scale agriculture in Taiwan, or areas with a similar climate to Taiwan’s, with a convenient and usable layer house with a water-pad system that considers the need for energy-saving and intelligent agriculture. It provides solutions for small-scale farming in terms of industrial upgrade and reduced energy consumption in the era of global warming and climate change.

A limitation of this study is that the humidity control design has yet to be implemented in commercial layer houses. The main reason for this is that farm operators are concerned about affecting current egg production. Furthermore, we know that increasing wind speed can help reduce perceived temperature. However, the effect of wind speed on the animal’s perceived temperature cannot be directly obtained from the animal’s response, which is research that needs to be continued in the future.

## Figures and Tables

**Figure 1 bioengineering-10-00452-f001:**
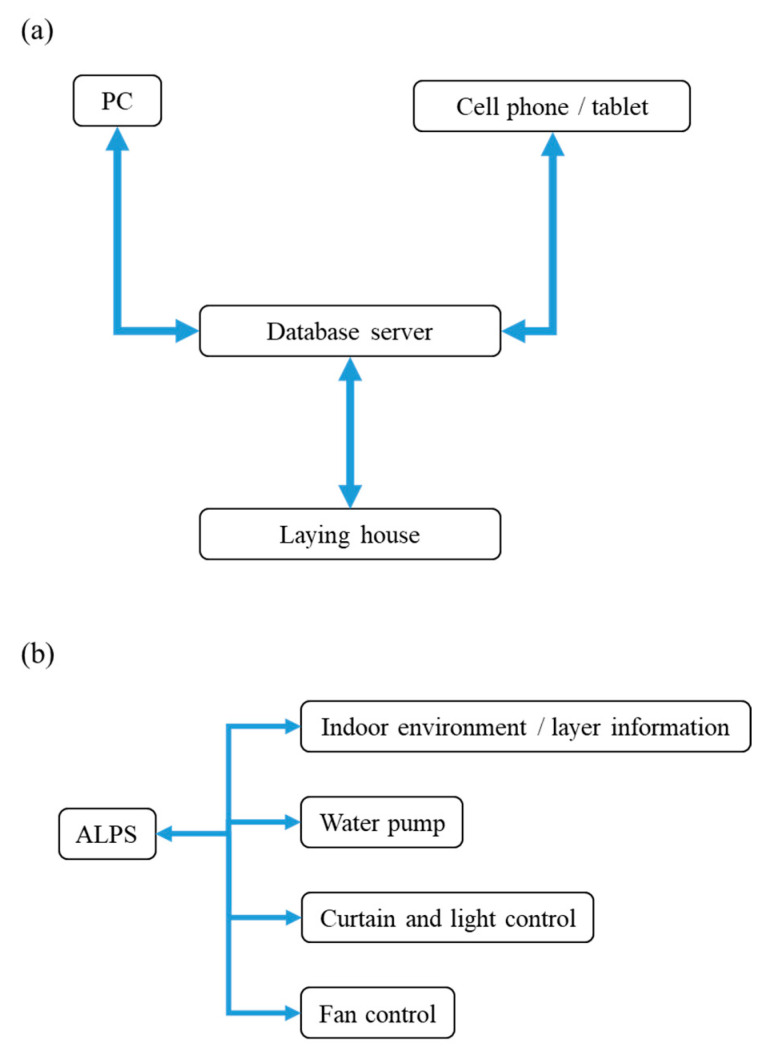
(**a**) Workflow of the Artificial Intelligence Layer Production System (ALPS). (**b**) Operation interface of ALPS.

**Figure 2 bioengineering-10-00452-f002:**
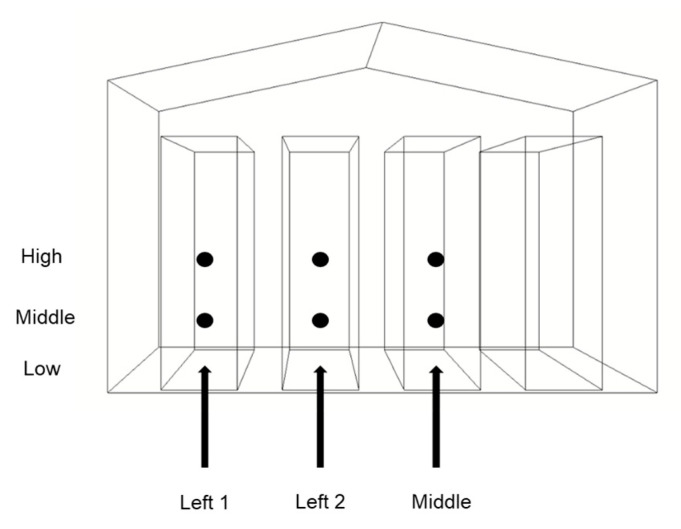
Temperature monitoring in water-pad layer house, Dalin, Chiayi.

**Figure 3 bioengineering-10-00452-f003:**
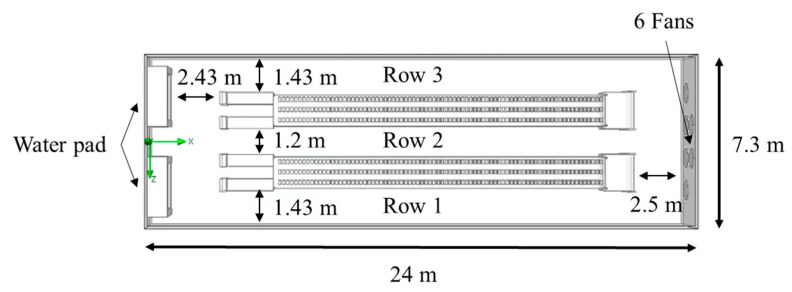
Dimensional drawing of layer house.

**Figure 4 bioengineering-10-00452-f004:**
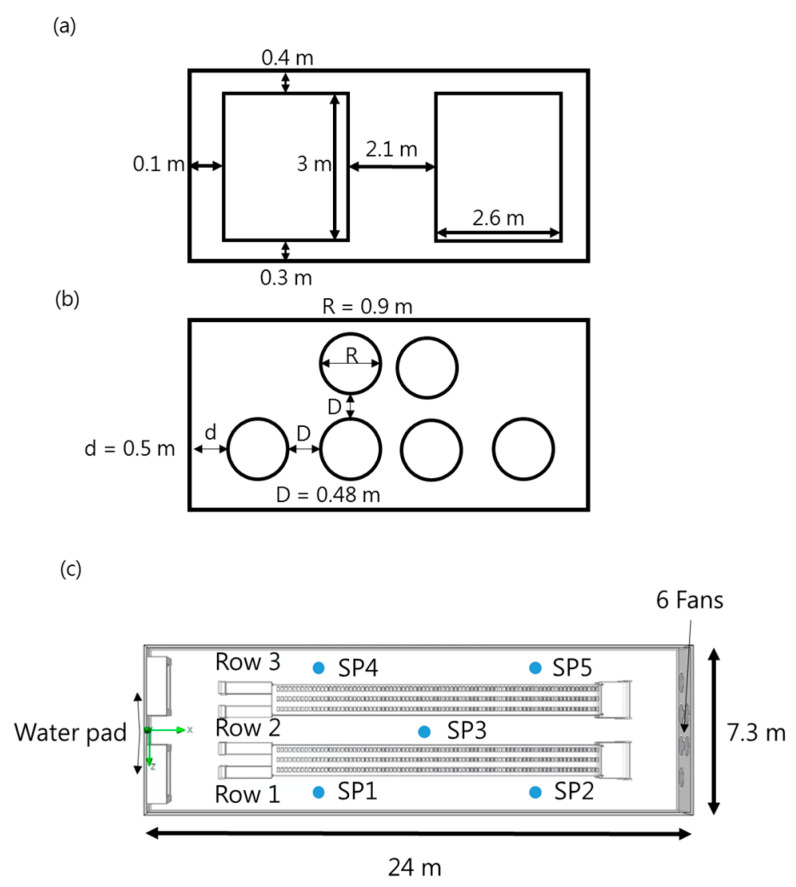
(**a**) Dimensional drawing of the water-pad system; (**b**) Dimensional drawing of the fan; (**c**) Location of measuring points SP1–SP5.

**Figure 5 bioengineering-10-00452-f005:**
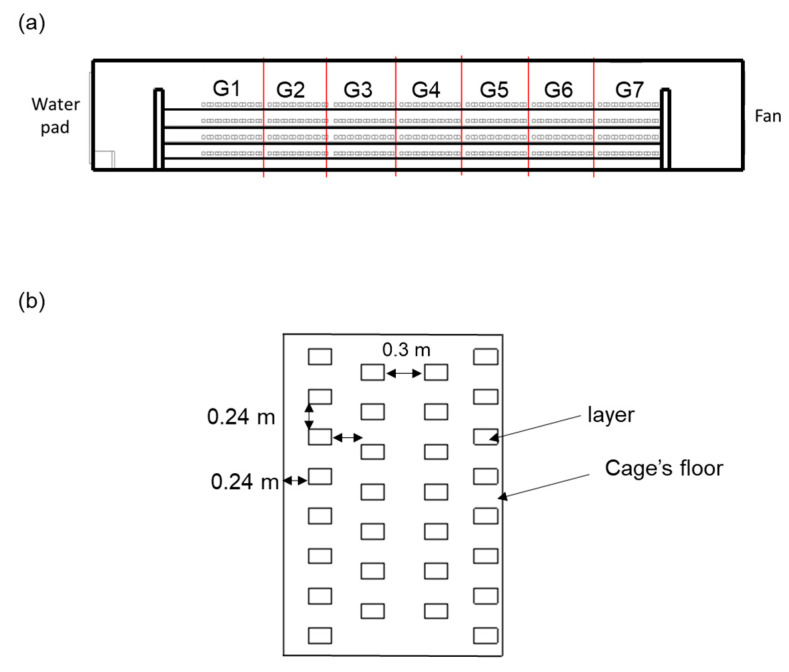
Information of cage. (**a**) Group of cages; (**b**) Schematic diagram of hens’ arrangement.

**Figure 6 bioengineering-10-00452-f006:**
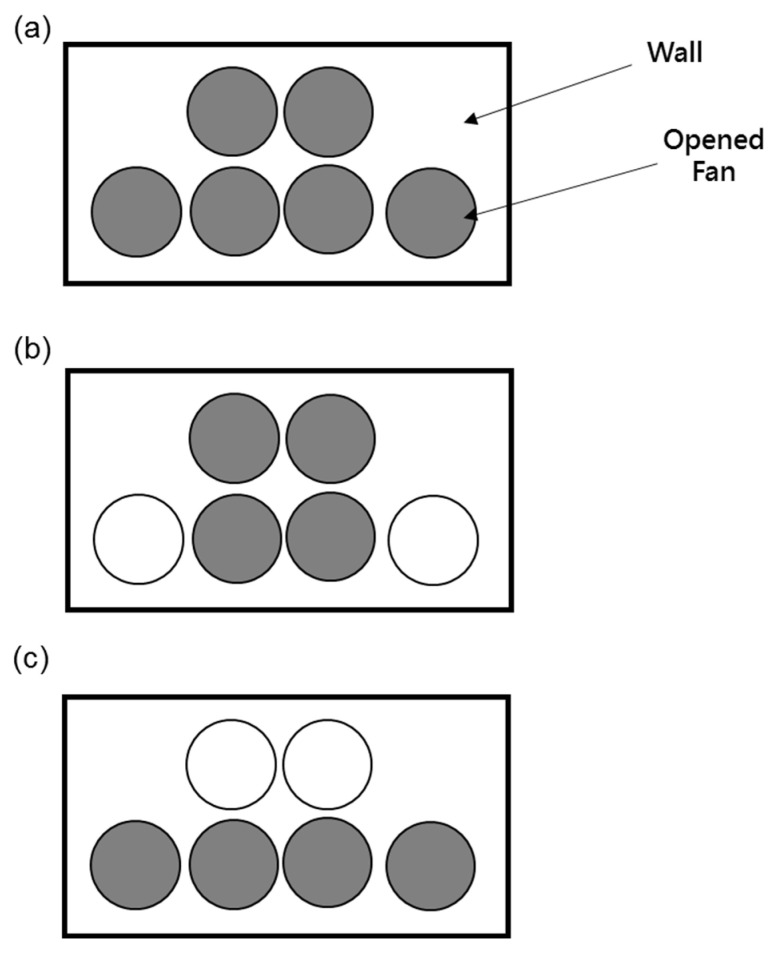
Three fan modes. (**a**) All fans were active; (**b**) four central fans were active; (**c**) four fans below were active.

**Figure 7 bioengineering-10-00452-f007:**
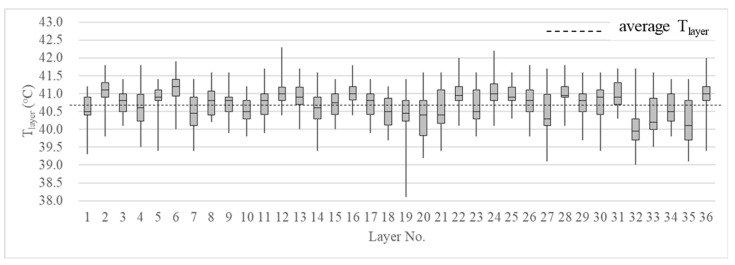
Body surface temperature of hens.

**Figure 8 bioengineering-10-00452-f008:**
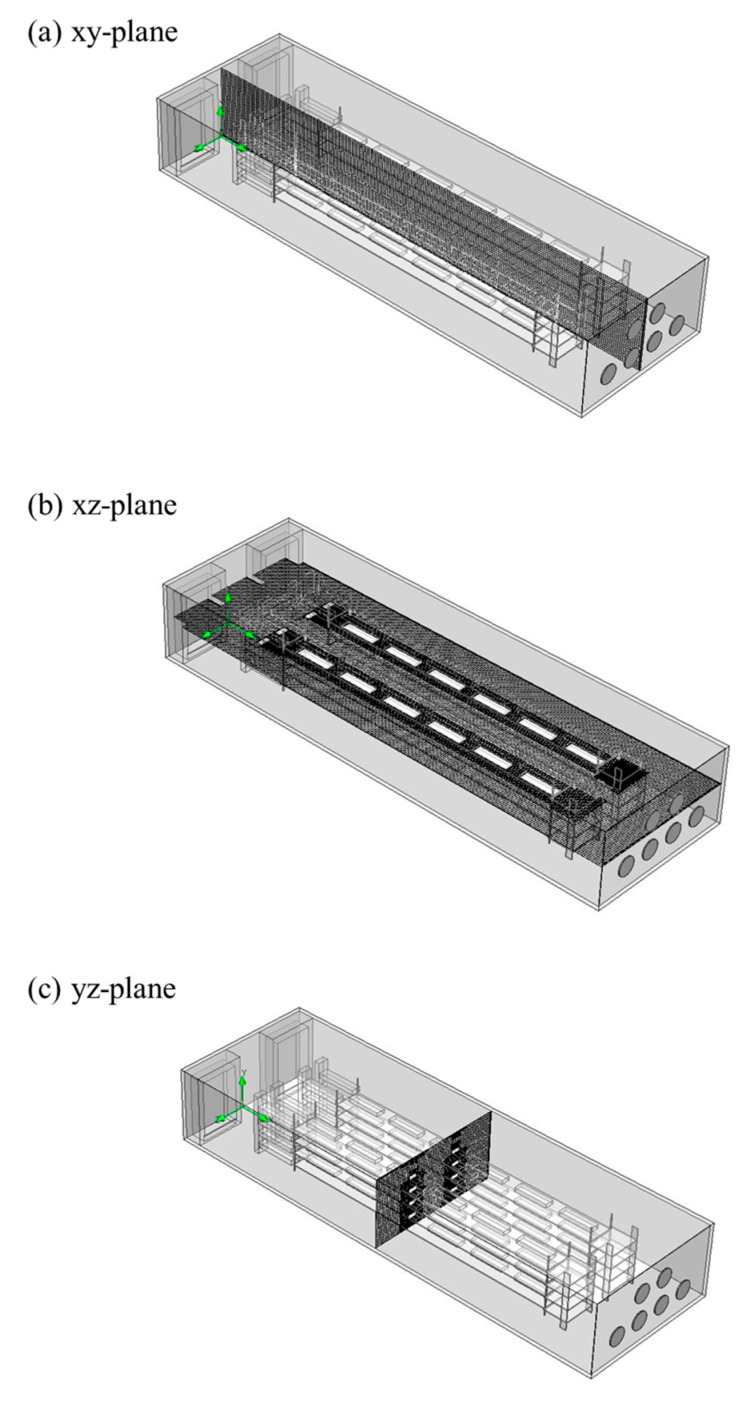
Mesh of layer house. (**a**) xy-plane; (**b**) xz-plane; (**c**) yz-plane.

**Figure 9 bioengineering-10-00452-f009:**
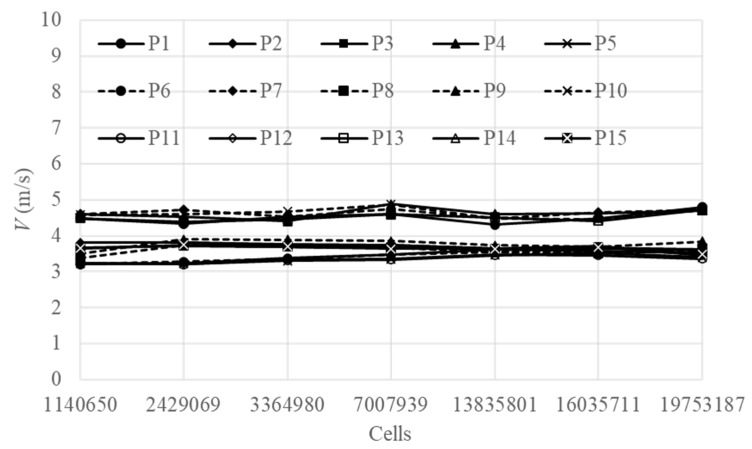
Grid independence of CFD Model.

**Figure 10 bioengineering-10-00452-f010:**
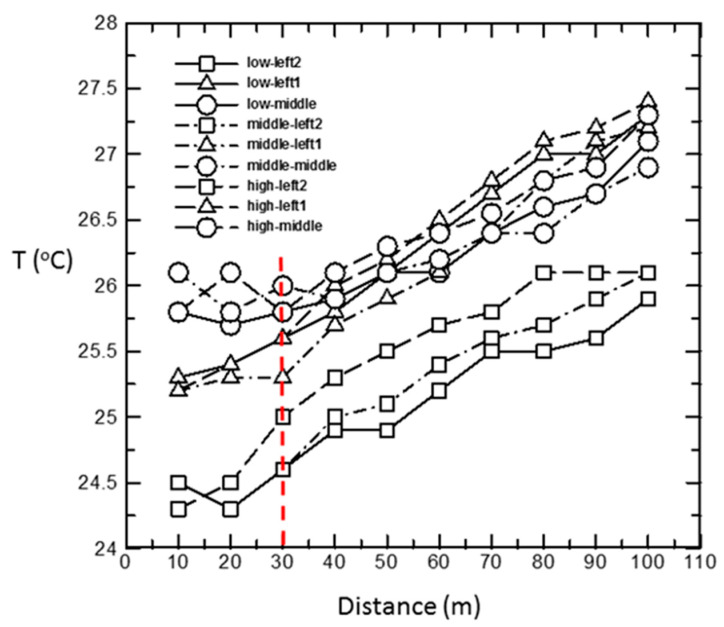
Temperature distribution in water-pad layer house, Dalin, Chiayi.

**Figure 11 bioengineering-10-00452-f011:**
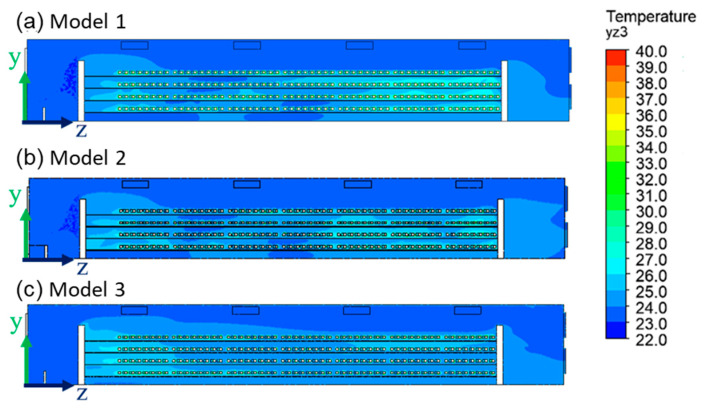
Temperature distribution of the three models’ yz plane of x = 2.6 m, at V = 6 m/s.

**Figure 12 bioengineering-10-00452-f012:**
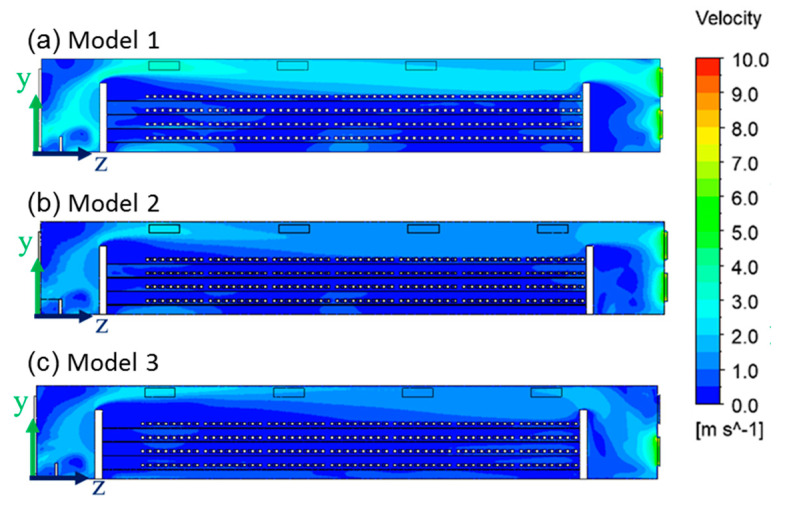
Velocity distribution of the three models’ yz plane of x = 2.6 m, at v = 6 m/s.

**Table 1 bioengineering-10-00452-t001:** ALPS in traditional layer house in Jiuru, Pingtung.

Item	Before	After
Number	7307→7146 (−161)	7146→7045 (−101)
Recording period (week)	4	4
Average egg weight (g)	48.64	47.86
Daily production rate (%)	69.73	73.24
Daily death rate (%)	0.086	0.041
Feed conversion ratio	2.38 ± 0.25	2.39 ± 0.08

**Table 2 bioengineering-10-00452-t002:** ALPS in water-pad layer house, Dalin, Chiayi.

Item	Before	After
Number	58,620→57,286 (−1,334)	56,086→55,330 (−756)
Recording period (week)	4	4
Average egg weight (g)	60.91	60.85
Daily production rate (%)	70.8 (54,344~53,001)	92.1 (51,001~52,117)
Daily death rate (%)	0.082	0.049
Feed conversion ratio	2.46 ± 0.12	2.45 ± 0.22

**Table 3 bioengineering-10-00452-t003:** Temperature difference between filed measurement and CFD model.

*H* (m)	Location	*T*_m_ (°C)	*T*_cfd_ (°C)	Difference (%)
2	SP1	26.5	26.7	0.6
	SP2	28.2	27.2	3.8
	SP3	28.4	26.6	6.2
	SP4	27.3	26.6	2.3
	SP5	30.3	27.2	10.3
2.7	SP1	26.8	26.6	1.0
	SP2	28.9	27.0	6.7
	SP3	28.0	26.9	3.9
	SP4	26.7	26.6	0.5
	SP5	29.6	27.0	8.8

**Table 4 bioengineering-10-00452-t004:** Temperature in cages.

*v* (m/s)	Model	Temperature (°C)								
		G1	G2	G3	G4	G5	G6	G7	Difference between G1 and G7	Average
2	1	25.8	25.7	26.0	26.2	26.3	26.5	26.5	0.7	26.1
	2	25.5	25.5	25.8	26.3	26.8	27.1	26.9	1.4	26.3
	3	25.6	25.7	25.9	25.8	26.1	26.3	26.5	0.9	26.0
4	1	25.2	25.0	25.3	25.5	25.6	25.9	26.0	0.8	25.5
	2	25.3	25.1	25.6	25.7	25.9	26.1	26.2	0.9	25.7
	3	25.4	25.3	25.3	25.4	25.5	25.8	25.9	0.5	25.5
6	1	25.1	24.9	25.2	25.2	25.4	25.7	25.6	0.6	25.3
	2	25.2	25.0	25.1	25.4	25.7	25.7	25.9	0.7	25.4
	3	25.2	25.3	25.2	25.2	25.3	25.5	25.6	0.4	25.3

**Table 5 bioengineering-10-00452-t005:** THI of groups in models at *v* = 6 m/s.

Humidity	Model	G1	G2	G3	G4	G5	G6	G7	Average
80%	1	75.2	74.9	75.4	75.4	75.8	76.3	76.2	75.6
	2	75.4	75.1	75.3	75.7	76.2	76.3	76.7	75.8
	3	75.4	75.7	75.3	75.4	75.6	75.9	76.1	75.6
90%	1	76.2	75.9	76.3	76.4	76.8	77.3	77.2	76.6
	2	76.4	76.0	76.2	76.7	77.2	77.3	77.7	76.8
	3	76.4	76.7	76.3	76.4	76.6	76.9	77.1	76.6
100%	1	77.1	76.8	77.3	77.4	77.7	78.2	78.2	77.5
	2	77.4	77.0	77.2	77.7	78.2	78.3	78.6	77.7
	3	77.3	77.6	77.3	77.3	77.5	77.9	78.0	77.6

**Table 6 bioengineering-10-00452-t006:** THI of Model 3 in groups at *v* = 6 m/s.

Humidity	Model	G1	G2	G3	G4	G5	G6	G7	Average
40%	Model 3	71.1	71.3	71.1	71.1	71.3	71.6	71.7	71.3
50%	Model 3	72.3	72.5	72.2	72.3	72.4	72.8	72.9	72.5
60%	Model 3	73.3	73.6	73.3	73.3	73.5	73.9	74.0	73.6
70%	Model 3	74.4	74.7	74.3	74.4	74.6	74.9	75.0	74.6

## Data Availability

Not applicable.

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
