# Peer review of "Assessing Indoor Climate Control in a Water-Pad System for Small-Scale Agriculture in Taiwan: A CFD Study on Fan Modes"

_bioengineering, 2023, doi:10.3390/bioengineering10040452_

Round 1

Reviewer 1 Report

The authors presented numerical study on the Assessing of Indoor Climate Control in Water-Pad System for Small-Scale Agriculture in Taiwan.

The main quantitative results are to be mentioned is the abstract.

The introduction is relatively short and may be extended.

The novelty of the paper is to be clearly stated.

The boundary conditions are to be expressed mathematically.

How can you study the effect of the air humidity without considering the mass transfer equations?

The numerical method is to be detailed.

A figure presenting the used mesh is to be added.

A grid sensitivity test is to be performed.

What is the convergence criterion?

The authors considered steady state equations, thus no need to present the time dependent equations.

The energy equation in the solid domains is to be presented.

Have you considered the hens as a heat source or adiabatic ?

Have you considered the external walls as adiabatic?

How is the water-pad system treated numerically ? have you considered it as porous media? Have you considered the phase change …?

The 3D flow structure (streamlines) is to be presented.

The 3D and 2D temperature profiles are to be presented.

The paper is to be checked against misprints and grammatical mistakes.

Author Response

Reviewer 1

  1. The main quantitative results are to be mentioned is the abstract.

Response:

The abstract was revised. Please see the line 14-30.

  1. The introduction is relatively short and may be extended.

Response:

The introduction was revised. Please see line 36-83.

  1. The novelty of the paper is to be clearly stated.

Response:

The added part was in line 432-435.

  1. The boundary conditions are to be expressed mathematically.

Response:

The boundary conditions were expressed mathematically. Please see line 209-234.

  1. How can you study the effect of the air humidity without considering the mass transfer equations?

Response:

When studying air humidity, mass transfer equations are important to consider as they govern the transport of water vapor in the air. However, in some cases, it may be sufficient to measure the real conditions of air humidity without explicitly considering these equations. My research mainly focuses on indoor airflow rather than discussing the primary mechanism of moisture in the air. To a certain extent, I ignored the influence of the moisture transfer equation by directly measuring the humidity level of the indoor air and using it as a boundary condition for the simulation.

  1. The numerical method is to be detailed.

Response:

The extended content was added in Line 235-258.

  1. A figure presenting the used mesh is to be added.

Response:

The figure of mesh in the model was added. Please see line 258-260.

  1. A grid sensitivity test is to be performed.

Response:

The grid independence was shown in Figure 10. Please see line 262-264.

  1. What is the convergence criterion?

Response:

“…The grid independence results showed that for grid sizes above 13,835,801, the values of the 15 monitoring points reached convergence criterion of 10-3…” Please see line 255-257.

  1. The authors considered steady state equations, thus no need to present the time dependent equations.

Response:

The transient state forms were deleted. Please see Line 205-208.

  1. The energy equation in the solid domains is to be presented.

Response:

The manifestation of the energy equation in the model is the calculation of temperature, which has been revised and included in the boundary condition section. Please see line 209-234.

  1. Have you considered the hens as a heat source or adiabatic?

Response:

The hen is a warm-blooded animal, so we consider it as a heat source with a stable surface temperature, rather than being in an adiabatic state. Please see line 221-226.

  1. Have you considered the external walls as adiabatic?

Response:

The actual field has adiabatic walls. We measured the temperature on the walls in the layer house. We found the wall temperature inside the room is stable and not affected by the external environment. Therefore, we treated the wall temperature as a fixed temperature for calculations. Please see line 233-234.

  1. How is the water-pad system treated numerically? have you considered it as porous media? Have you considered the phase change …?

Response:

Because our research primarily focuses on indoor microclimate rather than the specific water pad system, then considering the water pad as a porous media or taking into account phase change may increase computational time and deviate from our main objective. In this case, we may choose to simplify the model by treating the water pad as a single heat transfer surface. This can help us obtain simulation results more quickly and better support our main research objectives.

  1. The 3D flow structure (streamlines) is to be presented.

Response:

We have tried a 3D flow to describe our study. However, we found the 3D flow was not a suitable method for observing the flow in a layer house. We considered that a 2D structure can present the flow details and characteristics more clearly, and that data analysis and comparison can be more easily performed.

  1. The 3D and 2D temperature profiles are to be presented.

Response:

2D temperature distribution was to be presented in Figure 12. We have tried a 3D temperature profile to describe our study. However, we found the 3D flow was not a suitable method for observing the temperature in a layer house. Please line 360-364.

  1. The paper is to be checked against misprints and grammatical mistakes.

Response:

The manuscript has been checked. Thank you for the comment.

Reviewer 2 Report

The authors have performed valuable research on layer production systems.  The study appears to be well designed and nicely executed.  The paper is broadly well prepared.

Overall 

Why is the paper only focused on Taiwan and small-scale agriculture.  The conclusions may have greater impact.

Table 1 

1.     Is there a way to include standard errors and statistical analysis?

Table 1 and 2

2.     I am confused by the age in days and recording period in weeks. 

Figure 9 

3.     The legend requires further explanation. 

4.     What is distance relative to?

Table 5

5.     The legend requires further explanation. 

6.     Why are there no standard errors and statistical analysis?

Results

Where is data on surface temperature in the birds?

Author Response

Reviewer 2

The authors have performed valuable research on layer production systems.  The study appears to be well designed and nicely executed.  The paper is broadly well prepared.

Overall 

Why is the paper only focused on Taiwan and small-scale agriculture.  The conclusions may have greater impact.

Response:

“Taiwan's agricultural economy is predominantly based on small-scale farming. The defining feature of this system is that many farming households own small plots of land and focus on cultivating crops such as fruits, vegetables, rice, and flowers. This model has many advantages, including the promotion of rural employment and higher income for farmers. It also helps to maintain the efficiency of land use, improve product quality, protect the environment, and preserve rural culture and community cohesion. However, small-scale farming also faces many challenges, such as a lack of resources, insufficient technological advancement, and market instability. Therefore, Taiwan needs to implement appropriate policies to support small-scale farming, improve agricultural productivity and competitiveness, and promote sustainable agricultural development.” We added this paragraph in Introduction of the manuscript. Please see line 65-74.

Table 1 

  1. Is there a way to include standard errors and statistical analysis?

Response:

Thank you very much for your comment. We can only provide the information that the manufacturer has provided, as they consider much of the information to be confidential. This means that we cannot obtain more detailed information. If there are similar research opportunities in the future, we will communicate with the manufacturer as much as possible to ensure we obtain more information.

Table 1 and 2

  1. I am confused by the age in days and recording period in weeks. 

Response:

We revised Table 1 and 2. We will adopt weeks as the standard form for recording data. Please see line 285-286 and line 298-299.

Figure 9 

  1. The legend requires further explanation. 

Response:

The legend explains that the sampling location is at the position mentioned in Figure 3. Please see line 143-144.

  1. What is distance relative to?

Response:

The distance is about 10 meters between each point.

Table 5

  1. The legend requires further explanation. 

Response:

The legend explains that the sampling location is at the position mentioned in Figure 6(a).

  1. Why are there no standard errors and statistical analysis?

Response:

Data in Table 5 were calculated by the CFD in the study so that there were no standard errors and statistical analysis.

Results

Where is data on surface temperature in the birds?

Response:

Figure 8 shows the surface temperature of the birds.

Reviewer 3 Report

this article is well written. Some minor comments to be adressed are:

1. Abstract is recommended to contain introduction, objective, methods, result and short conclusion. The introduction part is missing, please add this part in the abstract.

2. The introduction is interesting and the authors have highlighted the state of the art. I suggest authors to highlight the novelty this study.

3. It is nice if authors add reference to Figure 1, if necessary

4. For treacibility, the methods should be accompanied with Reference

5. In table 1, the statistical test is recommended to be used to know the significant difference between before and after. The authors can use independent t-test for this case.

6. In discussion, it is better if authors compare their results with the previous study

Author Response

Reviewer 3

this article is well written. Some minor comments to be addressed are:

  1. Abstract is recommended to contain introduction, objective, methods, result and short conclusion. The introduction part is missing, please add this part in the abstract.

Response:

“Heat stress poses a significant challenge to egg production in layer hens. High temperatures can disrupt the physiological functions of these birds, leading to reduced egg production and lower egg quality.” We added the section in the part of Introduction. Please see line 14-16.

  1. The introduction is interesting and the authors have highlighted the state of the art. I suggest authors to highlight the novelty this study.

Response:

“The novelty in this study is the use of computational fluid dynamics (CFD) to investigate the impact of fan mode on indoor climate control in the water-pad system. This represents a new approach to studying the indoor climate of the water-pad system and provides insights into how to optimize the system for improved temperature distribution.” We added the section in Conclusion of the manuscript. Please see line 432-435.

  1. It is nice if authors add reference to Figure 1, if necessary

Response:

Thank you for your suggestion. We would like to clarify that Figure 1 illustrates the workflow of the Artificial Intelligent Layer Production System (ALPS). This research represents the first academic paper to showcase ALPS. Please see line 130-131.

  1. For treacibility, the methods should be accompanied with Reference

Response:

The manuscript's references to the methods have been listed in "Numerical Simulation" of "Material and Methods." Please see line 243-249.

  1. In table 1, the statistical test is recommended to be used to know the significant difference between before and after. The authors can use independent t-test for this case.

Response:

Thank you very much for your comment. We can only provide the information that the manufacturer has provided, as they consider much of the information to be confidential. This means that we cannot obtain more detailed information. If there are similar research opportunities in the future, we will communicate with the manufacturer as much as possible to ensure we obtain more information.

  1. In discussion, it is better if authors compare their results with the previous study

Response:

“Heat stress refers to the pressure that high temperature and humidity in the environment place on living organisms. High temperature and humidity hurt their productivity in the production process of laying hens. The perceived temperature generated by temperature and humidity is a component of heat stress assessment. As temperature and humidity increase, the perceived heat sensation of laying hens also increases, affecting their productivity. Therefore, to maintain the productivity and health status of laying hens, it is necessary to monitor and regulate the temperature and humidity in the environment, keeping them within a range that the hens can adapt to.” We added the section in “Discussion” of the manuscript. Please see line 392-398.

Round 2

Reviewer 1 Report

Accepted 

Reviewer 2 Report

The authors have done a good job revising the manuscript.